# Coexisting with the Life of Patients with Hemodialysis: Qualitative Meta-Synthesis Study of Life of Caregivers of Patients with Hemodialysis

**DOI:** 10.3390/ijerph19042163

**Published:** 2022-02-14

**Authors:** Eun-Young Kim, Ye-Na Lee

**Affiliations:** 1College of Nursing, Korea University, Seoul 02841, Korea; magic0614@korea.ac.kr; 2Department of Nursing, The University of Suwon, Hwaseong 18323, Korea

**Keywords:** renal dialysis, caregivers, systematic review, qualitative research

## Abstract

As the number of patients with hemodialysis (HD) continues to increase, so too does the number of people depending on caregivers. There is need to pay attention to the lives of caregivers of patients with HD, where sacrifices are forced. This study systematically reviewed and synthesized qualitative studies that explored the experiences of caregivers caring for patients with HD using the meta-synthesis method. We searched literature using four databases (i.e., PubMed, Excerpta Medica dataBASE, Cumulated Index to Nursing and Allied Health Literature, and Web of Science), and finally ten publications were selected. Four themes and nine subtopics were derived from analyzing and synthesizing the research results. The synthesized themes were: “bearing the burden of life as a caregiver,” “reconstructing life to maintain hemodialysis”, “the fading of caregiver’s own life,” and “effort to relieve the burden.” The results of this study can contribute to the development of interventional studies to improve the quality of life of HD patients. These studies provide an integrated and in-depth perspective on the experiences of caregivers who care for HD patients.

## 1. Introduction

As the number of patients with chronic renal failure increases globally, hemodialysis (HD) has been gaining popularity as a maintenance treatment approach [1]. Despite technical development and change, HD remains a challenging and difficult treatment for patients and requires support from caregivers at home or in hospitals [2]. Therefore, caregivers can experience many issues, including psychological, physical, and social pain [2,3].

Reasons for their pain include the need for provision for long-term treatment, the long-term course of disease, the possibility of HD-caused complications, new demands, and major changes in their lifestyle [3,4]. All of these factors can cause an extreme level of tension and lead to a collapse of the patient care structure. In addition, psychological problems caused by reduced income, sexual disability, absence from work, forced leave of absence request, or early retirement to take care of the patient at home can lead to the personal lives of caregivers falling apart [5,6]. These issues ultimately affect the management of the patients, or early retirement to take care of the patient is a factor in the course of the disease [7]. Therefore, care for patients with HD should include care and support for the caregivers as well, and the understanding of caregivers on the part of healthcare professionals is essential in effective and continuous management of patients’ illnesses [8,9]. In particular, healthcare professionals are in a great position to assist patients and caregivers in adapting to changes, building a positive relationship, and enhancing strategies to help manage the changes caused by HD [10]. Accordingly, healthcare professionals must first understand caregivers’ experiences before providing support. It is difficult for quantitative research to comprehensively review and analyze individual experiences, but qualitative research is necessary to understand the practical difficulties and effects of these issues to determine their degree of influence as opposed to quantitative indicators [11]. Although quantitative studies outnumber qualitative studies, a comprehensive analysis of the latter provides meaningful data by allowing for an in-depth understanding of the situational context and personal experiences of each individual [12]. A qualitative meta-synthesis method extracts common major concepts that could not have been identified through individual studies. It usefully provides a novel interpretation of the results by deriving and integrating the core concepts while respecting the uniqueness of each study through a deep interpretation of individual research results [12].

The review of some previous meta-synthesis studies can be difficult when such studies include both healthcare professionals and patients with HD, peritoneal dialysis, and organ transplantation as participants. This is because the diversity in participants makes it difficult to understand the experience of caregivers for HD patients [13,14]. In addition, very few studies have limited their participants to mothers or parents, which are a special subgroup of caregivers. This subsequently complicates the establishment of a general understanding of the caregiver status [15,16]. Therefore, there is a strong need for a review of qualitative studies using a set criterion for caregivers who care for HD patients to provide basic data for educational purposes or policies for caregivers of HD patients.

This study aimed to explore the experiences of nurses through analyzing and synthesizing the findings of qualitative studies on the experiences of caregivers of HD patients. The objective of this study is to establish an in-depth understanding of common experiences among caregivers as well as common findings from questionnaires by comparing, analyzing, interpreting, and synthesizing the findings of qualitative studies that examined the experiences of caregivers who care for HD patients. By presenting a conceptual framework that extends beyond the simple collection of the primary results of qualitative research, the researchers intend to provide fresh insight into the experience of caregivers caring for HD patients. Based on the results, this study intends to provide basic data for improving the quality of nursing and education for caregivers caring for HD patients in the future and ultimately offer meaningful data for improving the quality of life of both HD patients and caregivers. Therefore, the purpose of this study is to systematically review and synthesize the qualitative basis for understanding the experiences of caregivers for HD patients.

## 2. Methods

### 2.1. Study Design

This qualitative meta-synthesis study analyzed and synthesized the findings of qualitative studies and investigated the experiences of caregivers caring for HD patients. This study adopted a meta-ethnography method that used a line-of-argument synthesis method suggested by Noblit and Hare (1988) [17]. This synthesis method re-conceptualizes the existing concept and consists of the following seven steps: (1) getting started, (2) deciding what is relevant to the initial topic of interest, (3) reading the studies, (4) determining how the studies are related, (5) translating the studies into one another, (6) synthesizing translations, and (7) expressing the synthesis. This meta-ethnographic systematic review was registered with the International Prospective Register of Systematic Reviews (PROSPERO): CRD42021282512. For conducting based on the Enhancing Transparency in Reporting the synthesis of Qualitative research statement (ENTREQ) guideline, this study included five domains of COREQ.

### 2.2. Data Collection Method

#### 2.2.1. Literature Research and Selection

In this study, a literature selection process was performed according to PRISMA’s systematic literature protocol. The researchers used four (4) electronic search databases, Pub//med, Excerpta Medica dataBASE, Cumulated Index to Nursing and Allied Health Literature, and Web of Sciences, as shown in Figure 1. The medical subject headings (MeSH) included “hemodialysis,” “caregiver experiences,” and “qualitative study.” The literature selection criteria are as follows: (1) caregivers as participants; (2) studies were published within the last ten years, from 1 August 2011, to 31 July 2021; (3) studies were published in English in peer-reviewed journals; and (4) participants were ≥18 years old. Exclusion criteria were (1) studies that did not fit the objective of this study, (2) non-English-language studies, (3) quantitative research studies, (4) studies with unclear or complex criteria for participant selection, (5) secondary analysis studies, and (6) studies for which no access existed to the full text. This review process identified 715 studies and eliminated 379 results that were repeated. The title and abstract of the remaining 336 studies were screened to determine if they met the inclusion criteria. Overall, 246 studies that did not meet the inclusion criteria were eliminated. Lastly, the full texts of 90 studies were reviewed, and inappropriate studies were eliminated based on the inclusion criteria. In conclusion, a total of 10 studies were selected for analysis.

#### 2.2.2. Quality Appraisal

The Critical Appraisal Skills Programme (CASP) was used to assess the quality of the 10 studies that made the final selection [18]. Designed by two researchers, CASP is a tool that assesses the credibility, trustworthiness, and rigor of qualitative studies. When the researchers differed, they came to an agreement after having discussions.

According to the results of a literature quality assessment, no studies were unclear in their description of whether participants met the inclusion criteria, and five studies did not state the relationship between the researcher and participants. In addition, all studies met the criteria with regard to the study objective, method, design, data collection method, ethical considerations, data analysis, results, and research implications. Given that most qualitative meta-synthesis studies assessed the quality of literature to understand the study, instead of conducting an evaluation, this study examined the quality of literature to adequately understand the selected studies. Therefore, for the article [No. 25] with low percentage, after discussion in order not to lose important first-order constructs, the researchers did not exclude the studies based on the results of the quality assessment (Table 1).

### 2.3. Data Analysis and Synthesis

Noblit and Hare’s (1988) analysis method integrates the core concepts of the qualitative studies by exploring the similarities and differences between the selected studies through comparative analysis [17]. In addition, this method is a synthesis process that explores and synthesizes the results from the qualitative studies. In the data extraction process, first-order constructs (direct quotation from the original study) and second-order constructs (the author’s conceptual interpretation of the original study) were extracted. The extracted data were used for analysis and synthesis. By comparing, contrasting, combining, and synthesizing the different studies, this method allows the researchers to understand the new perspective and central meaning of the phenomenon of interest and to gain an in-depth understanding of that phenomenon.

In this study, the selected studies were listed in the order of the year of publication from the most recent to the oldest. The researchers compared the concepts identified in the first literature with those identified in the second literature and then extracted the common concept and topic. By repeating this process, the broad concept was narrowed down to specific categories. Subsequently, the original data were re-analyzed to review the identified research results, and the reviewed research results were integrated into a common concept to derive a topic (third-order constructs) that could be considered a higher concept.

## 3. Results

As a result of the quality assessment, four studies had a 90% satisfaction rate in relation to CASP, five studies with 80%, and one study with 60% (Table 1).

In total, 133 caregivers participated in the 10 studies. Participants’ age range was 21–78 years, and the care period varied from two months to sixteen years. The types of caregivers included were spouses, adult children, parents, nieces, daughters-in-law, grandfathers, grand-daughters, sons-in-law, and medical staff (Table 2).

Four themes and nine sub-themes were derived from analyzing the final ten studies included (Table 3). The four themes are “bearing the burden of life as a caregiver”, “reconstructing life to maintain hemodialysis”, “the fading of caregiver’s own life”, and “effort to relieve the burden”, and the nine sub-themes are “ facing a variety of burdensome problems”, “living an unstable life between physical and mental stress”, “effort at home to maintain steady hemodialysis”, “resetting daily life around the patient”, “absence of the caregiver’s own life”, “weakening of relationships around caregivers”, “anxiety from an uncertain future”, “developing coping strategy”, and “seeking help to reduce the burden on themselves”.

### 3.1. Theme Ⅰ. Bearing the Burden of Life as a Caregiver

Family caregivers of HD patients were living with heavy demands. They faced a variety of problems that demanded their attention, and lived an unstable life with physical and mental stress.

#### 3.1.1. Sub-Theme 1. Facing a Variety of Burdensome Problems

Due to HD, patients felt a variety of emotions, and family caregivers had to deal with their emotions as well [19,20,21,22,23,24]. Caregivers had to care for excessively dependent patients [19,20,21,22,24,25,26]. However, caregivers still felt responsible for the patients’ lives [19,20,21,22,23,24,25,26,27]. As HD patients had to periodically undergo dialysis at the hospital, caregivers had to transport them to the hospital periodically [20,21,22,24,26,27]. In addition, they experienced a financial burden due to continuous HD and drugs required [20,22,24,27].

“The difficulty only when … like she cannot get up, have to carry her … Ya, to carry her, to bathe her.” (21, p.1227)

“Haha I know this is easier said than done … money is a very important thing. It will affect everything in our lives. Plus the fact that this is quite an expensive disease. I can only say as someone at the consumer end, more money is better” (21, p.1228)

“These patients have lots of difficulties and miseries such as transfer to the hemodialysis ward, doing household tasks, handling the patient’s bad temper, suffering from insomnia and so on. What should I say? They all affect the caregivers’ soul and mind so that I am willing to die.” (23, p.196)

#### 3.1.2. Sub-Theme 2. Living an Unstable Life between Physical and Mental Stress

Caregivers had to agonize together with patients who felt pain [19,20,22,23,24]. Since their emotions were ignored [20,21,25,26,28], they experienced physical [20,22,24,26,28] and emotional exhaustion [20,22,26,27,28]. They felt confused when caring for patients due to the uncommon complexity of the patient’s disease [20,22,23,24,25,28]. In addition, caregivers felt guilty, thinking that they were at fault for the patient’s disease, and that they were unable to take care of the patient properly [22,23,24,25,26,27].

“Those are the days [when patient undergoes fistula de-clot at the hospital followed by dialysis at the clinic] that are really tiring. Those are the days that I just pray that she makes it through and she keeps her head up. I try to motivate her to keep her spirit up as well as mine.” (19, p.1365)

“It’s the many restrictions that make diet hard to follow. And sometimes, a news report may say that for example, avocado or bananas are good for people with diabetes. But she cannot eat it because of the calories and potassium? Yeah. I may be wrong, but my experience is that so many foods are difficult to follow. Actually, I’m not sure which (i.e., diabetes vs. ESRD) diet matters and has an impact.” (21, p.1227)

### 3.2. Theme Ⅱ. Reconstructing Life to Maintain Hemodialysis

For patients, HD is an essential treatment for maintaining life; therefore, caregivers reconstructed their lives to maintain HD for patients. In this study, caregivers did their best as a family to ensure that patients can maintain stable HD. In addition, routines were re-scheduled so that the patient could receive regular HD smoothly.

#### 3.2.1. Sub-Theme 3. Efforts at Home to Maintain Steady Hemodialysis

Caregivers were responsible for taking care of HD patients’ health even at home. For HD patients, the fistula, like a lifeline, had to be carefully managed [19,22,24]. It was necessary to closely observe the patients because there is a possibility of sudden death if not properly managed at home [19,24,26]. To manage difficult-to-manage chronic renal failure, patients had to be controlled to ensure that treatment guidelines were followed [19,20,23,24,25,26,27]. Management other than HD was performed at home and required the cooperation of the whole family [19,23,26,27,28].

“I must take care of the fistula on her hand. She should take her medications on time. She needs to drink a little water and not consume certain foods. In sum, I am always mentally engaged, and I always think about the things that she should or should not have and whether there is pressure on the fistula of her hand?” (26, p.256)

“After some sessions, he gained weight, so his nurse asked why he has not observed the water restriction. Well, summer is hot, and my child becomes thirsty. What can I do? How can I tell him not to drink water? I am his mother.“ (27, e23)

#### 3.2.2. Sub-Theme 4. Resetting Daily Life around the Patient

Caregivers tried to provide personalized care for patients [19,20,21,23,25,26,27], and had to make decisions related to patient care [19,24,26]. Prioritization had to done based on the patient’s needs [19,20,21,23,25,26,28]. Caregivers had to adjust their daily routines for smooth HD treatment for patients [19,22,23,24,26,27,28].

“... In hard conditions of caretaking, both planning and prioritization are important to me, that is, (to decide) which job is more important and valuable for my wife. In this way, the caring plan proceeds more comfortably and I’m more satisfied, as well...” (20, p.167)

“... His condition may get worse and time is gold for us. For this reason, I’ve written down all his requirements and planned and really coordinated all the jobs I do for him...” (20, p.167)

### 3.3. Theme Ⅲ. The Fading of Caregiver’s Own Life

As they continued to live their lives for patients, the quality of life of caregivers gradually decreased. They were not present in their lives, the relationships around them weakened, and they were anxious about their uncertain future.

#### 3.3.1. Sub-Theme 5. Absence of the Caregiver’s Own Life

Caregivers only cared about the health of the patients, while ignoring their own physical health [20,21,22,23,24,27,28]. As their thought process was always patient centered [19,20,21,25,26,27,28], they did not engage in hobbies or leisure activities [24,25,26,27], and perceived that their quality of life had deteriorated [19,20,25,26,27,28].

“I was the only one that really cared for her and I’ve been helping her. I really care for her; I don’t want nothing to happen to her. There’s nothing to do but to help her. “ (19, p.1366)

“She didn’t go for dialysis and she’s at home, my mind is not at peace at work also. I’m afraid ‘cause she’s alone. For me, looking after someone who’s sick comes with a lot of problems. Want to go out is a problem. ‘Cause as caregivers, when we’re out our minds are not at peace too. We’ll be thinking… scared if my wife falls, scared she’s that” (21, p.1231)

#### 3.3.2. Sub-Theme 6. Weakening of Relationships around Caregivers

As caregivers led their lives centered on the patient, they neglected human relationships around them. For caregivers who took care of their spouses, as the sexual function of the patient’s spouse deteriorated, their marital relationship deteriorated because they were unable to lead a normal married life [22,27]. They neglected family members other than the patient [21,23,24,27,28], and family sacrifices followed to take care of the patient [20,22,23,24,26,27]. Caregivers found it difficult to maintain their social life [20,25,26,27], and the relationship of trust was broken by conflicting opinions with the medical staff and the feeling of being ignored by the medical staff [22,24,27].

“My wife is on dialysis for about 5 years. In the first and second years, we had approximately no problems. But this disease destroys the dialysis person’s sexual feelings and severely affects the patient’s spouse, while underlying nervous tensions.” (23, p.196)

“I come here and sit for four hours. No one speaks with me. Nurses only connect the machine to my child without asking, ‘Are you okay?’ I like having someone to speak” (27, p.e22)

#### 3.3.3. Sub-Theme 7. Anxiety from an Uncertain Future

Caregivers experienced various uncertainties and felt anxious about their future as well. They experienced the patient’s unpredictable physical condition [19,20,24,25,27,28] and felt that their future was uncertain [20,22,24,25,26,27]. As caregivers faced unemployment owing to their caregiver burden, their futures were uncertain [20,22,25,26]. In addition, they felt as if there was no end to the treatment [21,22,23,24,25,27], and experienced the anxiety of being trapped in a bridle. Recognizing that a kidney transplant is the only way for a patient to be cured in an endlessly uncertain and unstable tunnel, they began to look forward to the kidney transplant [24,25,27].

“Cause like me, I got no maid, so I have to sacrifice my job. I have to stop my job for 7 years and take care of my mother.” (21, p.1230)

“The patient’s care process is lengthy. The caregiver should get up at 6 am in the morning and take the patient to a dialysis ward. He/she should spend 3–4 h over the patient’s bed in the hemodialysis ward. When the patient comes back home, the caregiver should massage his/her legs for half an hour and check whether he/she has any bleeding. This process is repeated three times a week. I certainly assert that caregivers are permanently getting involved in the process of care.” (23, p.197)

“That’s the hardest part, how to plan for the future… Caring for someone who has cared for you; that’s the biggest transition.” (28, p.29)

### 3.4. Theme Ⅳ. Effort to Relieve the Burden

Caregivers made various efforts to relieve the burden of demands placed on them. They developed coping strategies to find their own life amongst their caregiving responsibilities, and tried to find help to reduce their burden.

#### 3.4.1. Sub-Theme 8. Developing Coping Strategies

Using a variety of methods, caregivers tried to live their own lives. They learnt how to live their lives through the experiences of acquaintances in similar situations [23,27,28]. They attained useful information from various resources [23,27,28] and tried to find personal time [25,26]. They tried to live their own life separately from the life of a caregiver [25,27], and tried to have a hobby [25,26,28].

“... Everyday I’m [searching] on the internet and in books to learn something new and to be able to overcome the problems rather than allow them to defeat me...” (20, p.165)

“I kick box for one thing…hitting and kicking something as hard as you can helps a whole lot to deal with the frustration.”, “Now I get on the treadmill and run for 25 to 30 min. It feels great.” (28, p.29)

#### 3.4.2. Sub-Theme 9. Seeking Help to Reduce the Burden on Themselves

As the caregivers realized that it was difficult to manage all the demands on their own, they wanted to seek direct help. They also depended on religion to relieve their anxiety and stress levels [19,21,23]. In addition, they tried to find and share their concerns with others [19,23,26,28] and sought a variety of resources to help them [19,20,23,25,26,28].

“... In these times, if you don’t shout, no one will come to help you. You should always say what you’re doing so that others will come to help you. At home, in the family and in the community...” (20, p.165)

“Your love and your innate beliefs do not tell you that the patient is your mother, father, brother or spouse, but it is your innate belief and love that make you serve them with all your heart without ever tiring. I believe that such service is a belief and a value, and I greatly respect it.” (22, p.86)

### 3.5. Third-Order Synthesis: Understanding the Experiences of Caregivers Caring for Hemodialysis Patients

As shown in Figure 2, we developed a synthesized conceptual model of the experience of a caregiver caring for patients with HD. The four themes revealed in this study were not separated from each other, but could be expressed as organically connected.

Caregivers go through “bearing the burden of life as a caregiver”, “reconstructing life to maintain hemodialysis”, and “the fading of caregiver’s own life”, and “effort to relieve the burden” was experienced in turn. These efforts again became the strength to “bearing the burden of life as a caregiver”. As such, it was found that these processes have a structure that is organically connected and circulated.

## 4. Discussion

The life of caregivers who care for patients with hemodialysis is closely related to the life of the patient, but receives less attention compared to the patient’s life. There is a need to recognize the life of caregivers as important, as it can affect the quality of care provided to the patient. The integrated results of this study are helpful in understanding the experience of caregivers of patients with HD from the point of view of caregivers themselves. The needs of caregivers caring for patients with HD are continuously increasing, and research on ways to improve their quality of life should be conducted in various fields. In addition, both patients and caregivers must constantly adapt to their circumstances and update their knowledge and skills in disease management.

According to this study, caregivers have various burdens and live with physical and mental stress. This result supports previous studies that reveal that caregivers taking care of HD patients live with various demands and stresses [29,30]. However, although the caregivers themselves are aware of their stress and demands, we should pay attention to the fact that it can be ignored and taken for granted. Rather than taking the unilateral sacrifice of caregivers for granted and living with them, it is necessary to develop concrete ways to reduce the various demands and physical and mental stress. For this, variable studies on factors that obstruct and enhance the quality of life of caregivers should be conducted, and various interventional research developments that can improve the quality of life of caregivers should be developed based on this evidence. This will improve not only the caregiver’s life but also the patient’s quality of life, based on previous studies that support the notion that the quality of life of caregivers affects the quality of life of patients [31,32].

Caregivers reconstruct their lives for the hemodialysis of their patients. This means ignoring the caregiver’s own life and leading a patient-centered life. They have to constantly manage the patients at home, which requires effort from the caregivers, patients, and the rest of the family. A previous study found that families living with patients with HD experienced stress that was comparable to that experienced by the patients [26]. Although it is necessary to adjust the lives of the family members to some extent for the HD, we must consider the lives of the rest of the family. In this process of reorganization of life, it is necessary to continuously consult with the patient and the family living together so that the relationship between the family members does not deteriorate and a good relationship can be maintained. It is also necessary to develop and provide a relationship improvement program to maintain their relationships positively. In addition, medical staff should continue to provide education so that they can set the boundary between the life of the family and the life of the patient at home.

As they care for people with HD, the lives of caregivers become faded and they lose their identity. Their relationships are weakened and they live with anxiety due to an uncertain future. While caregivers live their patient-centered lives, their quality of life gradually deteriorates. Caregivers neglect their health and do not engage in leisure activities, leading to a deterioration in their quality of life. A previous study found that caregivers caring for HD patients developed various physical health problems [33], indicating a deterioration in the quality of life of caregivers; this finding has been supported by the previous studies [34]. Based on these problems, we need to recognize the need for caregivers’ independent lives; numerous efforts will be needed to enable caregivers to live their own lives. Rather than putting too much demand on one caregiver, cooperation among several people is required. In addition, various supports will be needed so that caregivers can enrich their lives. To realize this, national level assistance and policy discussions are needed to develop a diverse manpower pool that can help the caregivers caring for HD patients through discussions with experts.

In a patient-centered life, caregivers make efforts to reduce their burden. They develop various coping strategies to overcome difficult situations, and try to find various forms of help to relieve their burden. In this process, they try to obtain information through the experiences of acquaintances or various resources. Efforts are made to develop hobbies, while taking personal time away from life as a caregiver. They depend on their religion and find others with whom they can care for the sick. Medical staff can boost these efforts by connecting caregivers with self-help groups or community support resources.

This study provided a comprehensive and deeper understanding of the caregivers’ experiences of caring for HD patients, emphasizing the importance of developing customized interventions. These interventions will help the caregivers adapt to an independent life. This study can be usefully used to identify the difficulties they face and provide basic data for intervention studies that can solve them.

### Limitations

Although this study intended to include not only family caregivers but also a wide range of studies related to various caregivers in the review, we found that most of the participants in the included studies were family caregivers according to the research inclusion criteria. Therefore, we suggest the need to analyze the experiences according to the role of the caregiver by further subdividing the types of caregivers in the future.

## 5. Conclusions

In order to improve the quality of life of patients suffering from lifelong hemodialysis, it is essential to understand the suffering of their caregivers and to try to improve their quality of life. This study provides the basis for further research to improve the lives of caregivers. Based on this study, it is expected that various intervention programs for the lives of caregivers taking care of patients with HD will be developed in the future.

## Figures and Tables

**Figure 1 ijerph-19-02163-f001:**
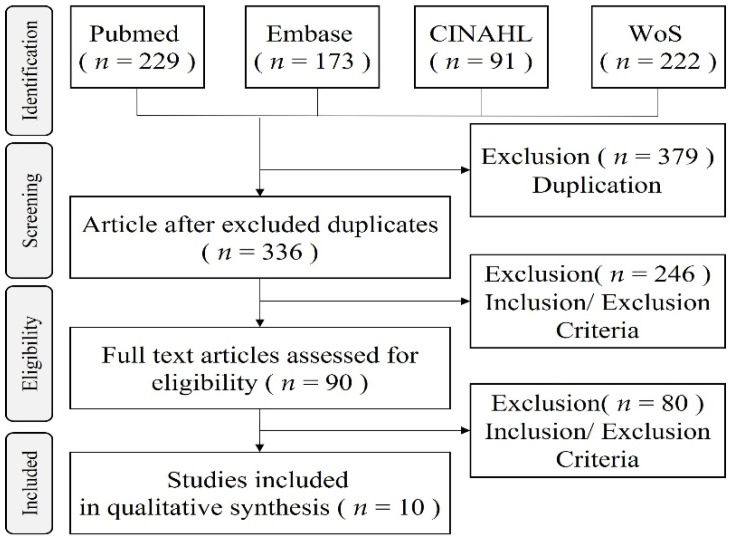
Flow chart of the systemic review in this study.

**Figure 2 ijerph-19-02163-f002:**
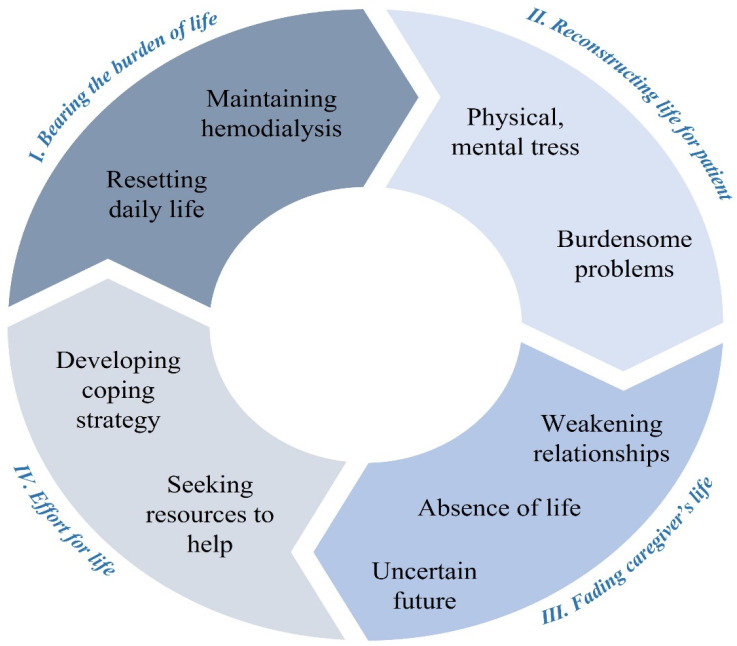
Conceptual model of the experience of a caregiver caring for patients with HD.

**Table 1 ijerph-19-02163-t001:** Quality assessment result of the critical appraisal screening program using studies included.

Article No.	19	20	21	22	23	24	25	26	27	28
Items										
1. Was there a clear statement of the aims of the research?	Y	Y	Y	Y	Y	Y	Y	N	Y	Y
2. Is a qualitative methodology appropriate?	Y	Y	Y	Y	Y	Y	Y	Y	Y	Y
3. Was the research design appropriate to address the aims of the research?	Y	Y	Y	Y	Y	Y	Y	Y	Y	Y
4. Was the recruitment strategy appropriate to the aims of the research?	Y	Y	Y	Y	N	Y	Y	Y	Y	Y
5. Was the data collected in a way that addressed the research issue?	Y	N	N	Y	Y	Y	N	Y	Y	Y
6. Has the relationship between researcher and participants been adequately considered?	N	N	Y	Y	Y	Y	N	Y	Y	N
7. Have ethical issues been taken into consideration?	Y	Y	Y	Y	Y	Y	N	Y	Y	Y
8. Was the data analysis sufficiently rigorous?	N	Y	N	N	Y	Y	N	N	N	N
9. Is there a clear statement of findings?	Y	Y	Y	Y	Y	Y	Y	Y	Y	Y
10. Was this research valuable?	Y	Y	Y	Y	Y	N	Y	Y	Y	Y
Percentage (%)	80	80	80	90	90	90	60	80	90	80

Y = Yes; N = No.

**Table 2 ijerph-19-02163-t002:** Summary of the studies included in the review.

Article No.	Author, (Year)	Sample Size (M:F)	Methodology	Age Range (Years)	Care Period	Relationship	Data Collection	Data Analysis	Key Findings
19	Amy O. Calvin et al. (2014)	18	Qualitative descriptive design	21–67 (Average: 42)	3–16 years	Spouse (7), Adult children (7), Parent (1), Sibling (1), Niece (1), Daughter-in-law (1)	Semi-structured interview	Glaserian approach	The overarching construct identified was one of Protection. Family members protect patients by ▪ Sharing Burdens Normalizing Life Personalizing Care.
20	Ahmad Ali Eslami et al. (2016)	20 (6:14)	Descriptive exploratory design	Average: 45	Not reported	Spouse (8), Daughter (6), Others (4)	Unstructured interview	Thematic analysis	▪ Help-seeking skills: Information gathering and continuous learning, Attempt to make others understand the situation, Seeking cooperation and assistance Self-nurturing skills: Enduring interaction, Adornment, Recreation and spirit renewal Skills in time management: Purposeful planning, Organization and prioritization Skills in stress management: Problem-focused coping, Role Modelling
21	Vanessa Y.W. Lee, et al. (2016)	20 (5:15)	Qualitative method	Average: 54.2 ± 12.6	Average: 7.1 ± 5.3 years	Son (1), Husband (4), Wife (10), Daughter (4), Mother (1)	In-depth interview	Inductive thematic analysis	▪ Challenges of caregivers: Diet and erratic appetite, Emotional management and interpersonal conflict, Mobility Limited resources: Poor knowledge and understanding; low perceived competence, Financial constraints, Lack of social support Impact of caregiving: Physical well-being, Empowerment status
22	Shahriar Salehi-tali et al. (2018)	16 (8:8)	Qualitative study	25–70 (Average: 41.5)	3–11 years (Average: 7.34 years)	Father (1), Mother (2), Spouse (4), Boy (2), Daughter (1), Grandfather (1), Patient (1), Physician (1), Nurse (1), Social worker (1)	Semi-structured interview	Conventional content analysis	Commitment to care ▪ Cultural and religious constructs : Patient’s dignity, Inherent love to care, Religious beliefs Sense of responsibility : Sense of duty, Marital commitment Self-restraint : Obeying the patient, Patience, Tolerance, Satisfaction with the situation, Accepting the situation Satisfactory caring : Satisfaction received from efforts, Determined in care
23	Shahriar Salehitali et al. (2018)	16 (8:8)	Qualitative research	25–70	3–11 years	Spouse (4), Son (2), Mother (3), Father (2), Daughter (2), Grandfather (1), Nurse (1), Doctor (1)	Semi-structured interview and observations	Content analysis	Progressive exhaustion ▪ Care challenges Psychological vulnerabilities The chronic nature of care Care in the shade
24	Abbas Ebadi et al. (2018)	19 (8:11)	Qualitative study	27–78 (Average: 42.16 ± 48)	46.33 ± 97.24 months (6–84)	Parent (2), Daughter (4), Son (3), Spouse (6)	In-depth interviews	Content analysis	Suspended life pattern ▪ Imbalance between caregiving and life : Compulsive compliance, Suspension, and deferral of roles, Conflicts between leisure time and caregiving, Disruptions in occupational affairs, Caregivers’ time limits Ambiguity in life status : Fear and hope, Caregivers’ satisfaction with life depending on care recipient’s condition.
25	Dian Sari et al. (2018)	7	Descriptive phenomenology	Not reported	Not reported	Not reported	In-depth interviews	Colaizzi technique	▪ families’ response to childcare families’ coping strategies the impact of childcare for families family support families’ perceptions of changes in children undergoing hemodialysis therapy
26	Tayebeh Pourghaznein et al. (2018)	11	hermeneutic phenomenology (step1–2)	23–51 (Average: 38)	2 months–8 years	Mother (11)	Semi-structured interview	hermeneutic phenomenology (step 3–6)	Immersion in an ocean of psychological tension ▪ Bewilderment between hope and despair Endless concerns Agony and sorrow Sense of being ignored
27	Tayebe Pourghaznein et al. (2018)	11	hermeneutic phenomenology (step1–2)	23–51 (Average: 38.00 ± 9.00)	2 months–8 years	Mother (11)	Semi-structured interview	hermeneutic phenomenology (step 3–6)	▪ Mothers enclosed by childcare Emotional and psychological tension Acceptance and contrivance The entire family being a victim of a sick child Self-devotion.
28	Christine Turner, and Patricia Finch-Guthrie(2020)	6	descriptive phenomenology	29–56	3.36 years (2 months–10 years)	Granddaughter (1), Daughter (3), Wife (1), Son-in-law (1)	Semi-structured	Giorgi method	▪ Caregiving is hard work : Organizing care requires planning, Co-morbid conditions compound care needs, A different challenge every day, Transportation is ongoing challenge Caregiving is stressful. : Compromised health of caregiver, An unbalanced life, An indeterminate timeline Caregivers need a support system. : Shared decision-making, Sharing the workload Caregiving is reciprocal. : Being an advocate Quality of life changes. : Family members’ life changes, Caregivers’ life changes Emotional responses to caregiving.

**Table 3 ijerph-19-02163-t003:** Family caregivers’ experiences of caring for patients with hemodialysis.

Key Concepts from First and Second-Order Constructs	Sub-Themes	Themes
Dealing with the patient’s emotions The patient’s excessive dependence Responsibility for the patient’s life Need to transport patients regularly Financial burden	Facing a variety of burdensome problems	I. Bearing the burden of life as a caregiver
The agony of having to watch the patient’s pain Ignoring their inner feelings Extreme physical fatigue Emotionally exhausted Confusion caused by the complexity of the patient’s disease Guilt for the patient	2. Living an unstable life between physical and mental stress	
Management of the patient’s fistula Close observation of a patient to prevent sudden death Controlling patients for treatment guidelines Cooperation of the whole family	3. Efforts at home to maintain steady hemodialysis	II. Reconstructing life to maintain hemodialysis
Personalizing care Decision-making about patient care Setting priorities based on patient’s need Adjusting daily life for hemodialysis	4. Resetting daily life around the patient	
Neglecting their physical health Thinking of everything as patient-centered Difficulty in leisure activities Decreased quality of life	5. Absence of the caregiver’s own life	III. The fading of caregiver’s own life
Deterioration of marital relationship Neglecting other family members Sacrifice of other families Difficulty in social life Feeling ignored by medical staff	6. Weakening of relationships around caregivers	
Unpredictable patient’s physical condition Uncertainty about caregiver’s future Disconnected employment Endless treatment process Expectations for a kidney transplant	7. Anxiety from an uncertain future	
Learning from the experiences of acquaintances Information from diverse resources Having personal time Being separated from the caregivers’ life Development of hobbies	8. Developing coping strategy	IV. Effort to relieve the burden
Dependence on religion Sharing the burdens with others Finding of resources to get help	9. Seeking help to reduce the burden on themselves	

## Data Availability

Not applicable.

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
