# Peer review of "Coexisting with the Life of Patients with Hemodialysis: Qualitative Meta-Synthesis Study of Life of Caregivers of Patients with Hemodialysis"

_ijerph, 2022, doi:10.3390/ijerph19042163_

Round 1

Reviewer 1 Report

Dear authors,

thank you for allowing me to view your interesting study. From the point of view of relevance, I think it is important. However, below you will find my suggestions for improving the article from a methodological point of view.

Kind regards

-----------------------------

Regarding critical appraisal, I would like to know the reason for the decision to exclude low quality articles. Don't you think you have lost important first-order constructs?

The reporting of findings is not smooth. I suggest anticipating findings with an overarching statement section explaining your results.

This study seems more like a meta-synthesis than a meta-synthesis. I think it might be useful to raise your analytical level to a more conceptual and interpretive level. 

Author Response

Response to Reviewers’ Comments regarding the manuscript entitled “Coexisting with the life of patients with hemodialysis : Qualitative meta-synthesis study of life of caregivers of patients with hemodialysis

We appreciate the very helpful comments on the initial draft. The manuscript has been strengthened significantly with your guidance.

We have tried our best to make revisions as reviewer comments have mentioned. We rewrote or rephrase to improve clarity and added to explain details about some vague points. To make it easier for you to follow how changes were made, we have revised the manuscript in red color.

Please consider the following revisions according to reviewers’ comments.

Thank you once again for all of your advice

Reviewer 1
Q1. Regarding critical appraisal, I would like to know the reason for the decision to exclude low quality articles. Don't you think you have ?

  • Author’s response : We agree with the reviewer's opinion. We included all studies without excluding them. Regarding critical appraisal was used for the purpose of identifying the studies. The researchers added and described more details.

Q2. The reporting of findings is not smooth. I suggest anticipating findings with an overarching statement section explaining your results.

  • Author’s response : We agree with the reviewer's opinion. The researchers additionally described the findings presented in the study earlier in the results section.

Q3. This study seems more like a meta-synthesis than a meta-synthesis. I think it might be useful to raise your analytical level to a more conceptual and interpretive level.

  • Author’s response : We agree with your concerns. We have rewritten our conceptual model (Figure 2), and provided an additional explanation for this. We hope this can give you a richer interpretation.

Reviewer 2 Report

The subject of your paper is relevant and fills a gap in the literature. Your methods deserve more detailed explanation, and the results could be presented more clearly.

Methods:

  • please give more details how you followed the ENTREQ guidelines (line 89-91).
  • why did you keep A7 in the analysis (tabel 1: only 60%).
  • please give more details how you applied Noblit and Hare's analysis method, especially in integrating the concepts (see also comments on results) (lines 130-137). Will this also explain the selection of particular quotes in the results?

Results:

  • in Table 2, last column: why are there different types of bullit markers in A4-A6 and A8. Please straigthen out the outline in this column (not centralised)
  • Table 3:
    • it is visually unclear which first-order constructs (first column) belong to which second-order constructs (second column) --> options are to include a clear white line, or perhaps a dashed line between  groups of first-order constructs.
    • in the third third-order construct: please add "of" in "The fading of caregiver's own life"
    • Why do you add "voluntary" to the fourth third-order constructs; no explanation whatsoever and this is not self-ecplaining from the perspective of caregiver who feels obliged to help his/her spouce.
    • Minor detail: there are some typo's in de list of A-references in the first column.
  • I find the set-up of the result a bit to simplistic and you leave the interpretation to the reader. You first explain which first-order constructs belong to a second-orde construct. And than you just list quotes. But then it becomes unclear which quote belongs to which first-order construct, and there is no storyline to the messages these particular quotes convey to get a meaningful interpretation to the second-order construct. So please rewrite all second halfs of all sections 3.1 to 3.4.
  • Section 3.5 is a very disappointing way to present your conceptual model. This is your novel end-result, please make more work out of this. It could also help to add between brackets your Latin numbering of the third-order constructs I tot IV. Als explain the reasoning behind the dashed arrow to combine I and II, besides their own direct arrows to III. Minors (repeated): whu voluntary? fading of caregiver's live
  • Discussion:
    • you mostly discuss  your results on the level of first- and second-order constructs, but it is more important to do that on third-level constructs and to embed the whole conceptual model in existing literature
    • most parts could be stated in present time. Only if you refer to A-references you may write in past time.
    • it would add meaning when you could extend the text with directions for future research and practical implications, on the level of your end-result i.e. the conceptual model. This would also provide a more proper place for your ideas about coustomizing interventions (now part of 5. Conclusions, but totally new there).
    • I really need a discussion about "voluntary".
  • Conclusions are not real conclusions, please rewrite.

Author Response

Response to Reviewers’ Comments regarding the manuscript entitled “Coexisting with the life of patients with hemodialysis : Qualitative meta-synthesis study of life of caregivers of patients with hemodialysis

We appreciate the very helpful comments on the initial draft. The manuscript has been strengthened significantly with your guidance.

We have tried our best to make revisions as reviewer comments have mentioned. We rewrote or rephrase to improve clarity and added to explain details about some vague points. To make it easier for you to follow how changes were made, we have revised the manuscript in red color.

Please consider the following revisions according to reviewers’ comments.

Thank you once again for all of your advice

[Methods]

Q1. Please give more details how you followed the ENTREQ guidelines (line 89-91). why did you keep A7 in the analysis (tabel 1: only 60%).

  • Author’s response : We accept your questions and agree that our explanations were not sufficient. More details have been added to the Method section. In addition, the authors assessed the quality of literature to understand the study; instead of conducting an evaluation, this study examined the quality of literature to understand the selected studies adequately. Therefore, for the A7 study with a low percentage, after enough discussion to avoid losing critical first-order constructs, the researchers did not exclude the studies based on the quality assessment results. These details also has been added.

Q2. Please give more details on how you applied Noblit and Hare's analysis method, especially in integrating the concepts (see also comments on results) (lines 130-137). Will this also explain the selection of particular quotes in the results?

  • Author’s response : We accept your questions and agree that our explanations were not sufficient. More details have been added to the Data analysis and synthesis section.

[Results]

Q3. In Table 2, last column: why are there different types of bullit markers in A4-A6 and A8. Please straigthen out the outline in this column (not centralised)

  • Author’s response : We have arranged Table 2 clearer and rearranged it to be united.

Q4. Table 3: it is visually unclear which first-order constructs (first column) belong to which secondorder constructs (second column) –-> options are to include a clear white line, or perhaps a dashed line between groups of first-order constructs.

  • Authors’ response : We have made the arrangement of Table 3 clearer and rearranged it so that it can be visually confirmed at a glance.

Q5. In the third third-order construct: please add "of" in "The fading of caregiver's own life"

  • Authors’ response : Thanks for your grammatical correction. As you suggested, I modified it to "The fading of caregivers' own life"

Q6. Why do you add "voluntary" to the fourth third-order constructs; no explanation whatsoever and this is not self-ecplaining from the perspective of caregiver who feels obliged to help his/her spouce.

  • Authors’ response : We agree with the reviewer's comments. Through discussion, the researchers admitted that the word "voluntary" was an inappropriate expression, and deleted the word "voluntary" from the text.

Q7. Minor detail: there are some typo's in de list of A-references in the first column.

  • Authors’ response : Thanks for your correction. As you suggested, I modified it.

Q8. I find the set-up of the result a bit to simplistic and you leave the interpretation to the reader. You first explain which first-order constructs belong to a second-orde construct. And than you just list quotes. But then it becomes unclear which quote belongs to which first-order construct, and there is no storyline to the messages these particular quotes convey to get a meaningful interpretation to the second-order construct. So please rewrite all second halfs of all sections 3.1 to 3.4.

  • Authors’ response : We accept your questions and agree that our explanations were not sufficient. A brief description of the first-order construct and second-order construct has been added to the Method section. In addition, our researchers decided that it would be appropriate to change the sub-title to 'theme' and 'sub-theme' in the results after discussion. Thank you for your meaningful comments.
  • Revision : (Page 4) In the data extraction process, first-order constructs (direct quotation from the original study) and second-order constructs (the author's conceptual interpretation of the original study) were extracted. The extracted data were used for analysis and synthesis.

Q9. Section 3.5 is a very disappointing way to present your conceptual model. This is your novel end-result, please make more work out of this. It could also help to add between brackets your Latin numbering of the third-order constructs I tot IV. Als explain the reasoning behind the dashed arrow to combine I and II, besides their own direct arrows to III.

  • Author’s response : Acknowledging that our figure was not convincing, we redrawn them after discussion between researchers and added explanations. We look forward to seeing our pictures help your understanding.

Q11. Minors (repeated): whu voluntary? fading of caregiver's live

  • Author’s response : The term "voluntary" has been deleted throughout the text.

[Discussion]

Q12. you mostly discuss your results on the level of first- and second-order constructs, but it is more important to do that on third-level constructs and to embed the whole conceptual model in existing literature most parts could be stated in present time. Only if you refer to A-references you may write in past time.

  • Author’s response : Thanks for your comments. We restructured the discussion based on the third-order construct. In addition, the contents related to the third-order construct, not the contents of the literature, were described in the present tense.

Q13. it would add meaning when you could extend the text with directions for future research and practical implications, on the level of your end-result i.e. the conceptual model. This would also provide a more proper place for your ideas about coustomizing interventions (now part of 5. Conclusions, but totally new there).

  • Author’s response : Thanks for your comments. We agree that our conclusions are not appropriate, so we added content about future research directions to the ‘discussion’ section, and rewrote the ‘conclusion’.

I really need a discussion about "voluntary"

  • Author’s response : The term "voluntary" has been deleted throughout the text.

[Conclusions]

Q14. Conclusions are not real conclusions, please rewrite.

  • Author’s response : We rewrote the conclusion section.

Reviewer 3 Report

The study presents an interesting topic and a particular approach to the methodology of analysis. I think the results are relevant but I suggest some improvements for publication.

  • Line 30: Correct "long-t".
  • Line 127: Specify the meaning of Y - N as a footnote in Table 1.
  • Line 148: As written, it is understood that in each study the sample was composed of 133 caregivers, what seems unlikely. I understand authors would like to indicate that, in total, 133 caregivers participated in the 10 studies.
  • Table 2: Review the use of bullets in the key findings list and unify the criteria. Not understood correctly.
  • Line 138: The phrase doesn’t make sense. I guess the authors want to say something like "In this study, the selected studies were listed in the order of the year of publication from the most recent to the oldest".
  • Table 3: I recommend removing the bullets from the list for easy reading.

Author Response

Response to Reviewers’ Comments regarding the manuscript entitled “Coexisting with the life of patients with hemodialysis : Qualitative meta-synthesis study of life of caregivers of patients with hemodialysis

We appreciate the very helpful comments on the initial draft. The manuscript has been strengthened significantly with your guidance.

We have tried our best to make revisions as reviewer comments have mentioned. We rewrote or rephrase to improve clarity and added to explain details about some vague points. To make it easier for you to follow how changes were made, we have revised the manuscript in red color.

Please consider the following revisions according to reviewers’ comments.

Thank you once again for all of your advice

Q1. Line 30: Correct "long-t".

  • Author’s response : Thanks for your correction. We modified it.

Q2. Line 127: Specify the meaning of Y - N as a footnote in Table 1.

  • Author’s response : We specified that Y=yes and N=no below Table 1.

Q3. Line 148: As written, it is understood that in each study the sample was composed of 133 caregivers, what seems unlikely. I understand authors would like to indicate that, in total, 133 caregivers participated in the 10 studies.

  • Author’s response : Thank you so much for your meticulous correction. As you suggested, we revised it to "In total, 133 caregivers participated in the 10 studies."

Q4. Table 2: Review the use of bullets in the key findings list and unify the criteria. Not understood correctly.

  • Author’s response : We have arranged Table 2 clearer and rearranged it to be united.

Q5. Line 138: The phrase doesn’t make sense. I guess the authors want to say something like "In this study, the selected studies were listed in the order of the year of publication from the most recent to the oldest".

  • Authors’ response : Thanks for your correction. As you suggested, I modified it.

Q6. Table 3: I recommend removing the bullets from the list for easy reading.

  • Author’s response : Thanks for your advice, we removed the bullet from table 3. This modification improved the readability of Table 3.

Round 2

Reviewer 1 Report

Dear authors,

I find your study improved from the previous version. Thank you for following some of my suggestions. The method section is now clearly understood. The representation of results is clearer now.

Kind regards

Reviewer 2 Report

The authors have addressed the review comments adequately.